# Factors Associated with Students Meeting Components of Canada’s New 24-Hour Movement Guidelines over Time in the COMPASS Study

**DOI:** 10.3390/ijerph17155326

**Published:** 2020-07-24

**Authors:** M. Claire Buchan, Valerie Carson, Guy Faulkner, Wei Qian, Scott T. Leatherdale

**Affiliations:** 1School of Public Health and Health Systems, University of Waterloo, 200 University Ave W, Waterloo, ON N2L 3G1, Canada; wei.qian@uwaterloo.ca (W.Q.); sleatherdale@uwaterloo.ca (S.T.L.); 2Faculty of Kinesiology, Sport, and Recreation, University of Alberta, Edmonton, AB T6G 2H9, Canada; vlcarson@ualberta.ca; 3School of Kinesiology, University of British, Columbia, Vancouver, BC V6T 1Z4, Canada; guy.faulkner@ubc.ca

**Keywords:** physical activity, sedentary behaviour, sleep, guidelines, adolescent

## Abstract

This study aimed to determine if secondary school students are meeting the new Canadian 24-Hour Movement Guidelines (24-MG), as well as each individual recommendation (physical activity; sleep; sedentary behavior) within the 24-MG, and which student-level characteristics predict meeting the 24-MG, both cross-sectionally and longitudinally. This study is the first to examine longitudinal changes in students meeting the 24-MG, as well as student-level characteristics that were predictive of favourable shifts in movement patterns. Cross-sectional data were obtained for 11,793 grade 9 students across Canada as part of the COMPASS study. Of this sample, 3713 students provided linked follow-up data from grade 9 to grade 12. The probability of meeting the guidelines was modeled using two-level logistic regression analyses, adjusting for student-level co-variates and school clustering. Only 1.28% (*p* < 0.0001) of the sample met the overall 24-MG. Among grade 9 students, 35.9% (*p* < 0.0001), 50.8% (*p* < 0.0001), and 6.4% (*p* < 0.0001) were meeting the individual recommendations for physical activity, sleep, and screen time, respectively. Of those students, less than half were still meeting them by grade 12. Community sport participation was the only predictor of all three individual recommendations within the 24-MG. Longitudinal analyses found that community sport participation and parental support and encouragement were significantly associated with Grade 12 students starting to meet the physical activity and screen time recommendations, respectively, after having not met them in grade 9. Findings can be used to inform policy and public health practice, as well as to inform future research examining causal relationships between the variables.

## 1. Introduction

Youth health behaviours, especially physical activity (e.g., jogging, swimming, skipping), sedentary behaviour (e.g., recreational screen time), and sleep, are expected to track into adulthood [1]. Despite the associated risks, the majority of Canadian adolescents report insufficient levels of physical activity [2] and sleep [3], and excessive levels of sedentary behaviour [4]. Traditionally, physical activity, sleep, and sedentary behaviour recommendations have existed in distinct guidelines, suggesting the behaviours are independent of each other [5,6,7]. In 2016, the Canadian Society of Exercise Physiology developed new Canadian 24-Hour Movement Guidelines (24-MG) that integrate the three movement behaviours (i.e., physical activity, sedentary behaviour, sleep) to provide a more comprehensive outline of a 24 h period [8]. These guidelines were the first of their kind internationally, with Australia subsequently adopting similar recommendations [9]. More recently, Canada, Australia, New Zealand, South Africa, and the World Health Organization have developed 24-MG for children aged 3–5 [10,11,12]. These new guidelines highlight the codependent nature of movement behaviours and stress the importance of maintaining a balanced movement profile across all age groups. Compliance with the new 24-MG has been proposed to help adolescents establish healthy lifestyle patterns that lay the proper foundations for adulthood; when individuals allocate more time to physical activity and sleeping, there is less time that remains to engage in sedentary behaviours, in turn resulting in a more favourable movement profile [13].

The 24-MG includes four components, outlined as (1) SWEAT; (2) STEP; (3) SLEEP; and (4) SIT, involving daily recommendations for the following: (1) moderate-to-vigorous physical activity (MVPA), where youth accumulate at least 60 min per day of MVPA involving a variety of aerobic activities, and incorporate vigorous physical activities and muscle and bone strengthening activities each at least 3 days per week; (2) light physical activity, where youth accumulate several hours of a variety of structured and unstructured light physical activities, such as walking; (3) sleep, specifically 9–11 h from ages 5–13, and 8–10 h after age 14; and (4) limiting sedentary behaviour, with recommendations for no more than 2 h of recreational screen time per day [8].

Cross-sectional data demonstrate that between 1 and 6% of Canadian adolescents are meeting the overall 24-MG [13,14], with similar results seen in international samples [15,16]. However, when the recommendations are examined individually, only 24% of Canadian adolescents are accumulating a minimum of 60 min per day of MVPA [13,14]; only 68% of Canadian adolescents are getting adequate sleep [13,14]; and between 51 and 95% of Canadian adolescents are exceeding the sedentary behaviour guidelines [4,13,14]. Previous research indicates that high school students are not meeting these guidelines, but it remains unknown how these behaviour patterns evolve over the course of secondary school. Improving our understanding of adherence to the 24-MG and how it changes during secondary school will allow for more targeted efforts to increase, or at minimum, maintain these healthy behaviours across all students.

School-age children (i.e., 5–17 years) engaging in high levels of physical activity, with low sedentary activities, and adequate sleep have more favourable health measures than their inactive, sedentary, and/or sleep deprived counterparts [17]. During the high school period, there are many factors that influence an adolescent’s movement profile. Cross-sectional research shows that adolescents are more likely to be physically active if they are male [18,19], white [19,20], report binge drinking [19], and have parental support and encouragement [21,22,23]. Furthermore, participation in varsity, community, and intramural sports has been shown to impact an adolescent’s likelihood of being physically active [24]. Similar trends can be seen with sleep, as Canadian research suggests that students who obtained longer sleep durations were more likely to be male, white, and from higher socio-economic status [25]. Sedentary behaviours showed slightly different trends, with male sex and current smoking status being associated with higher levels of daily sedentary behaviour, corresponding to poorer outcomes [4]. These trends suggest that, while males are more likely to participate in physical activity and get more sleep, they are also more likely to spend greater amounts of time in front of screens. Previous research has outlined individual characteristics associated with meeting each distinct recommendation, however, predictors of meeting the overall 24-MG have yet to be identified. Additionally, individual characteristics that predict relevant shifts in behaviour patterns during adolescence are unknown and need to be explored.

Given the large portion of Canadian adolescents that report sedentary behaviours [4], and the natural interactions between movement behaviours, understanding how behaviour patterns shift during adolescence as well as identifying characteristics that may predict favourable shifts in these behaviour patterns could provide valuable insight for future policy and prevention efforts. This study aims to expand this knowledge with the following three objectives: (1) to determine if Canadian students in grade 9 (aged 13–14 years) are meeting the overall 24-MG and individual recommendations; (2) to investigate which student-level characteristics predict meeting the 24-MG among the total grade 9 sample; and (3) to evaluate changes in guideline status between grades 9 and 12, and which student-level characteristics predict these changes. The specific trends under investigation are those who continue to meet guidelines; those who transition from meeting guidelines in grade 9, but do not meet guidelines in grade 12; and those who transition from not meeting guidelines in grade 9 to meeting guidelines in grade 12.

## 2. Materials and Methods

### 2.1. Design

COMPASS (Cannabis use, Obesity, Mental health, Physical activity, Alcohol use, Smoking, Sedentary behaviour) is a prospective cohort study (2012/2021) designed to collect longitudinal data from a large sample of grade 9 to 12 secondary school students in Ontario and Alberta. Students were recruited within participating COMPASS schools using active-information passive-consent protocols. The current study uses cross-sectional data from grade 9 students in Year 2 (Y2: 2013/2014) and longitudinal data from grade 9 students in Y2 who also provided follow-up data in Year 5 (Y5: 2016/17), when they were in grade 12. The student-level COMPASS questionnaire (Cq) collects health behaviour data as described elsewhere [26]. Ethics committee approval was received from the University of Waterloo, Office of Research Ethics (ORE #17264). A full description of the COMPASS study methods is available in print or online [26] (http://www.compass.uwaterloo.ca/).

### 2.2. Participants

In Y2, 89 public, private, and catholic secondary schools (79 in Ontario; 10 in Alberta) participated in COMPASS, with 11,793 grade 9 students completing the questionnaire. Sixty-eight students were removed owing to missing data on sex, resulting in a Y2 sample of 11,725 grade 9 students. In Y5, 68 of the Y2 COMPASS schools participated (61 in Ontario; 7 in Alberta). Self-generated identification codes were used to link the within school data sets between Y2 to Y5, creating our longitudinal data set using the COMPASS data linkage procedure [27]. We linked data from 3816 eligible grade 9 students in Y2 with data available from 4005 grade 12 students to create a data set that included 3713 students tracked from grade 9 (baseline) to grade 12 (follow-up). An additional 56 students were removed owing to missing data on sex, resulting in a final Y2 to Y5 longitudinal sample of 3657 students. The main reasons for non-linkage included students transferring schools or dropping out, students on spare scheduled free/study periods or absent during data collection, or inaccurate data provided in the linkage measures.

### 2.3. Measures

#### 2.3.1. Outcome Variables

The outcome variable of interest was whether youth (i.e., ages 13–19 years) were meeting the SWEAT, SLEEP, and SIT components of the 24-MG. The Cq contains previously validated measures to determine if participating students are meeting the SWEAT, SLEEP, and SIT components of the 24-MG; Cq data are not available to measure the light physical activity (STEP) component. To assess the guideline recommendation for MVPA, students were asked the following: how many minutes of (a) moderate intensity and (b) hard intensity PA they engaged in on each of the last 7 days (in hours (0 to 4) and minutes (0, 15, 30, 45)); and, on how many days in the last 7 days did they do exercises to strengthen or tone their muscles (e.g., push-ups, sit-ups, weight training)? (0 to 7 days). Students were classified as meeting the MVPA recommendation if they had performed 60 min of MVPA daily and vigorous-intensity (hard) PA or strengthening activities on at least 3 of the last 7 days, otherwise they were classified as not meeting the recommendation. To assess whether students were meeting the guideline recommendation for sleep, they were asked to report how much time per day they usually spend sleeping (in hours (0 to 9) and minutes (0, 15, 30, 45)). Students were classified as meeting the sleep recommendation if they were 13 years of age and reported between 9 and 11 h of sleep a day or 14 years of age or older and reported between 8 and 10 h of sleep a day. To assess if students were meeting the guideline recommendation for sedentary behaviour, students were asked to report the average time per day (in hours (0 to 9) and minutes (0, 15, 30, 45)) that they spent in five different recreational sedentary behaviours: watching/streaming TV shows or movies; playing video/computer games; talking on the phone; surfing the internet; and texting, messaging, and emailing. It was not possible to identify students that accumulated more than 9 h and 45 min of sleep or screen time per day within our data. We calculated total screen time based on the sum of values for each of the five response categories reported. Students were classified as meeting the sedentary behaviour recommendation if they reported ≤2 h of recreational screen time per day. Conversely, students were classified as not meeting the guideline, and thus sedentary, if they exceeded the 2 h per day of recreational screen time. These measures have been previously validated; self-report behaviours were shown to be significantly correlated with accelerometer measurements and showed good test–retest reliability, although, consistent with other self-report measures, students do tend to overestimate the magnitude of their physical activity [28].

#### 2.3.2. Predictor Variables

Smoking (tobacco) was assessed by asking respondents (a) if they have ever smoked 100 or more whole cigarettes in their life; and (b) on how many of the last 30 days did they smoke one or more cigarettes? Consistent with previously validated measures of current smoking [29], students who reported ever smoking 100 cigarettes and any smoking in the previous 30 days were classified as current smokers.

Binge drinking alcohol was assessed by asking respondents to report how often they had five drinks of alcohol or more on one occasion in the past 12 months. Consistent with previous research [30], those who reported binge drinking once a month or more were classified as current binge drinkers.

In order to measure sports participation, students were asked to report the following: (a) if they participate in intramurals (before-school, noon hour, or after-school physical activities organized at school) (yes vs. no/not available); (b) if they participate in varsity sports (competitive sports teams that compete against other schools) (yes vs. no/not available); and, (c) if they participate in community sports (league or team sports outside of school) (yes vs. no/not available).

Students were also asked to report if their parents/guardians were supportive in being physically active (very supportive, supportive, unsupportive, very unsupportive) and if their parents/guardians encouraged physical activity (strongly encourage, encourage, do not encourage or discourage, discourage, strongly discourage). Students who reported that they were both supported (very supportive, supportive) and encouraged (strongly encourage, encourage) by their parents/guardians were classified as having parental support. The demographic correlates of sex (male, female), age (in years), and ethnicity were also reported in the Cq.

### 2.4. Analyses

All analyses were implemented in SAS 9.4. Using the grade 9 cross-sectional sample, we explored descriptive statistics for all variables by sex. We then performed a series of two-level logistic regression analyses (PROC GENMOD) to model the probability of meeting the 24-MG recommendations for SWEAT (Model 1), SLEEP (Model 2), and SIT (Model 3). Models were adjusted for student-level predictor variables, including age, sex, ethnicity, and other health behaviours [19]. School clustering was included in the model with an exchangeable correlation structure. The empirical estimation results were presented.

Using the Y2 to Y5 longitudinal sample, we explored changes in students reporting meeting the SWEAT, SLEEP, and SIT recommendations within the 24-MG from grade 9 to grade 12. McNemar’s test (PROC FREQ) was used to test if changes in students meeting the recommendations within the 24-MG over time are significant. Using the sample of respondents in the longitudinal sample who reported meeting a particular 24-MG in grade 9, we then performed a series of two-level logistic regression analyses to predict whether they were still meeting that guideline in grade 12 (SWEAT (Model 4), SLEEP (Model 5), and SIT (Model 6)), adjusting for student-level predictor variables and school-level clustering. We then used the sample of respondents in the longitudinal sample who reported not meeting a particular 24-MG recommendation in grade 9, and performed another series of two-level logistic regression analyses to predict whether they did meet that guideline in grade 12 (SWEAT (Model 7), SLEEP (Model 8), and SIT (Model 9)), adjusting for student-level predictor variables and school-level clustering. We used the Hosmer–Lemeshow test to check the goodness of fit for all the binary outcome models (PROC LOGISTIC), ignoring the correlation. With the exception of Model 8, all others with *p*-value > 0.05 suggest the binary model assumption is reasonable. Descriptive statistics are presented for the entire sample and by sex. The results from the two-level logistic regression analyses are presented as odds ratios with 95% confidence intervals. The results with an alpha level of <0.05 were deemed statistically significant.

## 3. Results

### 3.1. Descriptive Statistics: Year 2 Cross-Sectional Sample

Among the cross-sectional grade 9 sample, 51.3% (*n* = 6016) self-identified as male and 48.7% (*n* = 5709) as female. Of the 11,138 students included in the analysis, 1.28% (*n* = 143) met the overall 24-MG, consisting of the three behaviour recommendations included in this analysis. Among the sample of grade 9 students, 35.9% (*n* = 4000) met the SWEAT recommendation, 50.8% (*n* = 5933) met the SLEEP recommendation, and 6.4% (*n* = 753) met the SIT recommendation, independently. Descriptive statistics for the grade 9 cross-sectional sample are presented in Table 1.

Comparing stratified models, it can be seen that males were more likely than females to meet SWEAT and SLEEP, but less likely to meet SIT (Table 1). Males were significantly more likely than females to report participating in intramural, varsity, and community sports, as well as having parents who were supportive of physical activity. The prevalence of smoking and binge drinking was not significantly different between males and females.

### 3.2. Logistic Regression Models: Year 2 Cross-Sectional Sample

The results of the two-level logistic regression analyses identifying which student-level characteristics predict meeting the 24-MG recommendations in the cross-sectional grade 9 sample are presented in Table 2. Grade 9 students were more likely to meet the SWEAT recommendation (Model 1) if they were male; a current binge drinker; participated in intramural, varsity, or community sports; or if their parents supported physical activity. Grade 9 students were more likely to meet the SLEEP recommendation (Model 2) if they were male, participated in community sports, or if their parents supported physical activity, and they were less likely to meet the recommendation if they were a current binge drinker. Grade 9 students were more likely to meet the SIT recommendation (Model 3) if they participated in varsity sports or community sports, and were less likely to meet the recommendation if they were male or a current binge drinker. The results of the two-level logistic regression analyses identifying which student-level characteristics predict meeting the overall 24-MG in the entire grade 9 through 12 sample are presented in Table 3. The only characteristic predictive of Canadian high school students being more likely to meet the overall 24-MG was if they began participating in community sports after grade 9.

### 3.3. Changes in Students Meeting the 24-Hour Movement Guideline Recommendations: Longitudinal Sample

As shown in Table 4, there were significant changes in students meeting the recommendations within the 24-MG over time. In grade 9, 34.3% of students met SWEAT, but by grade 12, only 25.2% of students were meeting the recommendation. Among students who met SWEAT in grade 9, only 41.5% were still meeting the recommendation by grade 12. Conversely, among students who were not meeting SWEAT in grade 9, 16.6% were meeting the recommendation by grade 12.

In grade 9, 53.3% of students met SLEEP, but by grade 12, only 32.7% of students were meeting the recommendation. Among students who met SLEEP in grade 9, only 41.9% were still meeting the recommendation by grade 12. Conversely, among students who were not meeting SLEEP in grade 9, 22.2% were meeting the recommendation by grade 12.

In grade 9, 7.1% of students met SIT, but by grade 12, only 5.1% of students were meeting the recommendation. Among students who met SIT in grade 9, only 26.7% were still meeting the recommendation by grade 12. Conversely, among students who were not meeting SIT in grade 9, 3.5% were meeting the recommendation by grade 12.

### 3.4. Logistic Regression Models among Students Meeting the Recommendations in Grade 9: Longitudinal Sample

The results of the two-level logistic regression analyses to model the probability of meeting the 24-MG recommendations in grade 12 among students who were meeting the recommendations in grade 9 are shown in Table 5. Grade 12 students were more likely to continue meeting SWEAT (Model 4) if they were male, reported being a current binge drinker since grade 9, had participated in community sports since grade 9, or reported their parents have either supported physical activity since grade 9 or began supporting physical activity after grade 9. Grade 12 students were more likely to continue meeting SLEEP (Model 5) if they participated in intramural sports since grade 9. Grade 12 students were less likely to continue meeting SIT (Model 6) if they participated in intramural sports in grade 9, but stopped participating by grade 12.

### 3.5. Logistic Regression Models among Students Not Meeting the Recommendations in Grade 9: Longitudinal Sample

The results of the two-level logistic regression analyses to model the probability of meeting the 24-MG recommendations in grade 12 among students who were not meeting the recommendations in grade 9 are shown in Table 6. Grade 12 students were more likely to start meeting SWEAT (Model 7) if they were male, started binge drinking after grade 9, started participating in community sports after grade 9, participated in community sports in grade 9 but stopped by grade 12, or had participated in community sports since grade 9. Grade 12 students were more likely to start meeting SIT (Model 9) if their parents went from not being supportive of physical activity in grade 9 to being supportive by grade 12.

## 4. Discussion

The new Canadian 24-MG were designed to provide evidence-based behavioural benchmarks associated with optimal health if children and youth adhere to them. The guidelines also provide a basis for public health messaging and surveillance [31]. This study was the first to examine longitudinal changes in students meeting the 24-MG, as well as student-level characteristics that were predictive of favourable shifts in movement patterns. Findings can be used to inform policy and public health practice, as well as to inform future research examining causal relationships between the variables.

Among our sample of Canadian adolescents, only 1.28% of grade 9 students met the overall 24-MG, fewer than has been previously reported in both Canadian and international samples [13,15,16]. This discrepancy may be explained by our use of stringent criteria and 7-day recall, which may not be reflective of long-term behaviour [32]. A substantial number of students ceased to meet the individual components of the guidelines by the time they reached grade 12. Of those who met the individual recommendations in grade 9, under half were still meeting the MVPA and sleep recommendations and only one-quarter were still meeting the screen time recommendation in grade 12. These large reductions may be owing to a variety of factors associated with high school and adolescence. It has previously been shown that there is a decrease in sport participation in students over the course of high school [33]. It can be presumed that students face an increased workload and greater academic pressure as they progress through high school, potentially contributing to these patterns, as a result of less leisure time. It should be noted that physical education in Canada is not required in many provinces beyond grade 9 [34], which may also contribute to a decrease in within-school MVPA.

Unsurprisingly, participation in community, varsity, and intramural sports was positively associated with students meeting the MVPA recommendation [22,24]. Interestingly, only community sport participation was associated across all three individual recommendations within the 24-MG. Students who participated in community sports were significantly more likely to be physically active, sleep more, and spend less time engaging in sedentary behaviours. These findings were further substantiated by our analysis that demonstrated that community sport participation was the only predictor associated with meeting the overall 24-MG. Students who began playing community sports after grade 9 were nine and a half times more likely to meet the overall guidelines. Even among students who were not meeting the guidelines in grade 9, when they began participating in community sports, they were nearly three times as likely to meet the guidelines in grade 12. With the exception of some tournaments or out of town games, community sport participation in Canada typically occurs almost entirely outside of school hours [24]. It is thus reasonable to assume that youth who engage in these activities have less time to allocate to recreational screen time. It is also possible that students who participate in community sports are those who actively search for opportunities to be physically active and alternatives to sedentary behaviours. High school athletes have been shown to have more favourable sleep patterns than their non-athlete counterparts [35]. Students engaging in sports outside of school hours likely have more structured daily routines requiring set sleep schedules [35]. In addition, athletes may have a greater understanding of the importance of obtaining adequate sleep to allow for muscle recovery and the prevention of injuries or burnout [35]. These results provide compelling evidence to encourage sport participation among high school students both within and outside of the school context.

Parental support and encouragement was also seen to have a significant effect on sport participation. Students whose parents supported and encouraged physical activity participation were 29% and 27% more likely to meet the MVPA and sleep recommendation, respectively. Additionally, students who started receiving parental support and encouragement were over four times more likely to meet the screen time recommendation in grade 12 after having not met them in grade 9. This presents a valuable opportunity to engage parents and foster the skills necessary to encourage and support their child’s active behaviour. Compliance with the sedentary behaviour guideline was lowest among the three movement behaviours; if this were improved, it is possible that a larger proportion of adolescents would meet the overall 24-MG. These findings are consistent with research suggesting that social support is an influential factor in the likelihood of adolescents being physically active [24]. Parents who understand the importance of physical activity participation and limited screen time may instill these values in their children, ultimately leading to more favourable movement patterns. It is possible that children whose parents do not encourage physical activity participation may be less likely to participate in community or varsity sports as they often require parental consent. Previous research has shown that there is minimal parental support for the health behaviours included in 24-MG in Canada, with support for MVPA being the lowest of the three [36]. This research, in combination with the current findings, highlights the important role that parents can play in encouraging healthy movement behaviours in their children.

In accordance with previous research showing that youth sport participation is associated with alcohol use [37], current binge drinkers were found to be more physically active. Students who were current binge drinkers in grade 9 were 50% more likely to meet the MVPA recommendation compared with their non-binge drinking counterparts. Moreover, it was found that students who took up binge drinking in grade 12 were two times more likely to meet the MVPA recommendation in grade 12 after having not met them in grade 9. Considering that binge drinking has been tied to participation in team sports, these findings are not surprising [37,38]. It has been speculated that adolescents who engage in binge drinking are more interested in being physically active [39]. Sport participation is typically a social activity [40], and it is thus possible that adolescents who engage in sports may be more likely to attend social gatherings and partake in risky social behaviours [41], including binge drinking. These findings should not be used to discourage participation in team sports, but rather serve as an indication that substance use behaviours should be monitored among these students. Team sport contexts could be used as an opportunity to inform students of the harms associated with binge drinking [42].

As expected, movement behaviours differed significantly among males and females [22]. While males were more likely to be physically active and obtain more sleep, females were 30% more likely to meet the screen time recommendations. The differences in physical activity and sedentary behaviour found in this study are consistent with previous research indicating that males tend to be more physically active and are more likely to accumulate over 2 h per day in screen time than young women [4]. Surprisingly, this research found significant differences in sleep duration between sexes, whereas previous research on Canadian adolescents suggests that differences among men and women’s sleep durations were relatively minimal [43].

### Limitations

These findings must be considered in the context of the following limitations. First, the physical activity data used in this study were collected using self-report questionnaires. Self-report data often result in an overestimation of true physical activity levels and an underreporting of risky behaviors such as smoking or binge drinking. The questionnaires used in this study, however, have been previously validated as reliable measures of physical activity and sedentary behaviour [28]. Second, the nature of the questions included in the questionnaire prevented the investigation of two issues. We were not able to assess whether students were meeting the light physical activity recommendation. We were also not able to explore which specific sports are associated with binge drinking. Future research should examine substance use among different sports in high school. If we could pinpoint which sports increase an adolescents’ likelihood of substance use, efforts could be targeted to specific populations to increase results.

## 5. Conclusions

This analysis demonstrates a picture that is all too common across Canada—that youth are not obtaining sufficient physical activity and sleep and are spending extended periods of time engaging in sedentary activities. Just over 1% of the students included in the COMPASS study from 2013–2017 were meeting the overall 24-MG. Moreover, the number of youth meeting each recommendation reduced drastically over the course of high school. Students were more likely to meet the recommendations if they were participating in extracurricular sports, had sufficient parental support and encouragement, and engaged in binge drinking. This study identified that community sport participation and parental support of physical activity were particularly pertinent to students’ likelihood of meeting the 24-MG over time. Our findings highlight the critical need for new interventions and measures promoting sport participation and fostering support skills among parents to improve health and movement behaviours among future generations. Moving forward, a multi-sectoral approach targeting policies, environments, and schools should be developed to help improve health and movement behaviours among adolescents. In an attempt to maximize outcomes, efforts should be directed at both social and individual levels.

## Figures and Tables

**Table 1 ijerph-17-05326-t001:** Descriptive statistics for the cross-sectional COMPASS Year 2 (2013–2014) grade 9 student sample. BMI, body mass index.

	Female (*n* = 5709) %(*n*)	Male (*n* = 6016) %(*n*)	Total Sample (*n* = 11,138) % (*n*)	*p*-Value; df
Age				0.002; df = 5
13	3.7 (213)	3.7 (220)	3.7 (433)	
14	77.1 (4404)	75.0 (4511)	76.0 (8915)	
15	18.3 (1047)	19.9 (1198)	19.2 (2245)	
16	0.2 (9)	0.5 (28)	0.3 (37)	
17	0.1 (4)	0.1 (7)	0.1 (11)	
18	0.6 (32)	0.9 (52)	0.7 (84)	
Ethnicity				0.0037; df = 1
Other	17.6 (1007)	19.7 (1187)	18.7 (2194)	
White only	82.4 (4702)	80.3 (4829)	81.3 (9531)	
Current Smoker				0.409; df = 1
Yes	6.8 (381)	6.5 (377)	6.7 (758)	
No	93.2 (5182)	93.5 (5456)	93.4 (10,638)	
Current Binge Drinker				0.7047; df = 1
Yes	9.9 (565)	10.1 (607)	10.0 (1172)	
No	90.1 (5128)	89.9 (5382)	90.0 (10,510)	
Participates in Intramurals at School				<0.0001; df = 1
Yes	36.5 (2063)	40.4 (2391)	38.5 (4454)	
No	63.5 (3586)	59.6 (3521)	61.5 (7107)	
Participates in Varsity Sports at School				<0.0001; df = 1
Yes	39.3 (2177)	47.6 (2766)	43.6 (4943)	
No	60.7 (3366)	52.4 (3042)	56.5 (6408)	
Participates in Community Sports				<0.0001; df = 1
Yes	51.8 (2919)	62.9 (3707)	57.5 (6626)	
No	48.2 (2716)	37.1 (2186)	42.5 (4902)	
Parents are Supportive of Physical Activity				<0.0001; df = 1
Yes	73.5 (4126)	78.1 (4597)	75.9 (8723)	
No	26.5 (1486)	21.9 (1290)	24.1 (2776)	
Meeting the 24-Hour Movement Guideline				
SWEAT				<0.0001; df = 1
Yes	29.3 (1604)	42.4 (2396)	35.9 (4000)	
No	70.7 (3878)	57.6 (3258)	64.1 (7136)	
SLEEP				<0.0001; df = 1
Yes	48.0 (2734)	53.4 (3199)	50.8 (5933)	
No	52.0 (2961)	46.6 (2793)	49.2 (5754)	
SIT				0.0025; df = 1
Yes	7.1 (407)	5.8 (346)	6.4 (753)	
No	92.9 (5292)	94.2 (5654)	93.6 (10,946)	

**Table 2 ijerph-17-05326-t002:** Two-level logistic regression models examining the probability of meeting the different 24-Hour Movement Guideline (24-MG) recommendations within the cross-sectional COMPASS Year 2 (2013–2014) grade 9 student sample.

	MODEL 1: SWEAT OR (95% CI) *n* = 10,255	MODEL 2: SLEEP OR (95% CI) *n* = 10,760	MODEL 3: SIT OR (95% CI) *n* = 10,769
Intercept	0.20 (0.06–0.71) *	0.08 (0.02–0.27) **	0.03 (0.00–0.26) *
Sex			
Female (Ref)	1.00	1.00	1.00
Male	1.65 (1.49–1.82) **	1.19 (1.10–1.28) **	0.71 (0.61–0.83) **
Current Smoker			
No (Ref)	1.00	1.00	1.00
Yes	1.12 (0.92–1.36)	0.95 (0.80–1.13)	0.80 (0.54–1.19)
Current Binge Drinker			
No (Ref)	1.00	1.00	1.00
Yes	1.50 (1.27–1.77) **	0.85 (0.73–0.99)	0.50 (0.35–0.72)
Participates in Intramurals at School			
No (Ref)	1.00	1.00	1.00
Yes	1.23 (1.11–1.36) **	1.07 (0.97–1.18)	1.13 (0.97–1.32)
Participates in Varsity Sports at School			
No (Ref)	1.00	1.00	1.00
Yes	1.58 (1.44–1.74) **	1.10 (0.99–1.23)	1.33 (1.11–1.58) *
Participates in Community Sports			
No (Ref)	1.00	1.00	1.00
Yes	1.84 (1.62–2.08) **	1.15 (1.05–1.26) *	1.22 (1.01–1.46) *
Parents Support Physical Activity			
No (Ref)	1.00	1.00	1.00
Yes	1.29 (1.16–1.44) **	1.27 (1.15–1.40) **	1.15 (0.94–1.40)
*p*-value of Hosmer-Lemeshow test	0.7362	0.5470	0.6827

Note: Results are presented as odds ratios (OR (95% confidence interval)) adjusted for age and clustering by school. * *p* < 0.05; ** *p* < 0.0001.

**Table 3 ijerph-17-05326-t003:** Factors associated with meeting the 24-Hour Movement Guideline (24-MG) within the longitudinal COMPASS Year 2 to Year 5 student sample.

Parameter		OR (95% Confidence Interval)
Intercept		0.00 (0.00–0.43) *
Age		1.86 (0.69–4.98)
Sex	2	2.59 (0.88–7.57)
Current Smoker ^†^	0–1	0.63 (0.07–5.61)
Participates in Varsity Sports at School	0–1	0.82 (0.07–10.28)
	1–0	2.81 (0.40–19.64)
	1–1	3.16 (0.50–20.13)
Participates in Community Sports	0–1	9.56 (1.20–76.34) *
	1–0	0.72 (0.06–8.55)
	1–1	4.14 (0.72–23.91)
Participates in Intramurals at School	0–1	0.96 (0.17–5.50)
	1–0	0.28 (0.03–2.72)
	1–1	0.97 (0.21–4.46)
Parents Support Physical Activity	0–1	2.31 (0.22–24.10)
	1–0	0.42 (0.02–7.18)
	1–1	0.80 (0.09–7.01)
Current Binge Drinker ^‡^	0–1	0.42 (0.08–2.12)

Notes: 0 = no, 1 = yes; 0–0 = no in grade 9, no in grade 12; 0–1 = no in grade 9, yes in grade 12; 1–0 = yes in grade 9, no in grade 12; 1–1 = yes in grade 9, yes in grade 12. ^†^ For current smoker, there were no respondents with the 1–0 or 1–1 response option for the 24-MG. ^‡^ For current binge drinker, there were no respondents with the 1–0 or 1–1 response options for the 24-MG. * *p* < 0.05; OR, odds ratio.

**Table 4 ijerph-17-05326-t004:** Changes in students meeting the 24-Hour Movement Guideline recommendations over time in the longitudinal COMPASS Year 2 (2013–2014) to Year 5 (2016–2017) student sample.

**SWEAT**				
Grade 9	Grade 12			McNemar’s test
	Did Not Meet Recommendation	Met Recommendation	Total	df = 1, 89.9, *p* < 0.0001
Did Not Meet Recommendation	1814	362	2176	
Met Recommendation	666	472	1138	
Total	2480	834	3314	
**SLEEP**				
Grade 9	Grade 12			McNemar’s test
	Did Not Meet Recommendation	Met Recommendation	Total	df = 1, 372.5, *p* < 0.0001
Did Not Meet Recommendation	1322	377	1699	
Met Recommendation	1125	811	1936	
Total	2447	1188	3635	
**SIT**				
Grade 9	Grade 12			McNemar’s test
	Did Not Meet Recommendation	Met Recommendation	Total	df = 1, 16.4, *p* < 0.0001
Did Not Meet Recommendation	3264	118	3382	
Met Recommendation	189	69	258	
Total	3453	187	3640	

**Table 5 ijerph-17-05326-t005:** Factors associated with meeting the 24-Hour Movement Guideline components in grade 12 among students who were meeting the guideline in grade 9.

Parameter		Model 4: SWEAT *n* = 836	Model 5: SLEEP *n* = 1325	Model 6: SIT *n* = 185
		OR (95% CI)	OR (95% CI)	OR (95% CI)
Intercept		0.04 (0.00–5.19)	0.71 (0.01–34.81)	0.18 (0.00–4917.7)
Sex	Male	1.49 (1.10, 2.01) *	0.97 (0.77–1.22)	0.87 (0.41–1.86)
Current Smoker ^†^	0–1 vs. 0–0	1.36 (0.87–2.13)	1.17 (0.81–1.69)	0.65 (0.06–7.61)
	1–0 vs. 0–0	-	-	-
	1–1 vs. 0–0	0.46 (0.12–1.73)	1.22 (0.50–2.96)	-
Current Binge Drinker ^‡^	0–1 vs. 0–0	1.38 (0.95–2.00)	0.95 (0.70–1.29)	0.51 (0.16–1.60)
	1–0 vs. 0–0	1.47 (0.47-4.62)	0.67 (0.19–2.36)	-
	1–1 vs. 0–0	2.44 (1.01–5.91) *	0.95 (0.47–1.93)	-
Participates in Intramurals at School	0–1 vs. 0–0	0.76 (0.46–1.26)	1.23 (0.84–1.80)	0.41 (0.12–1.43)
	1–0 vs. 0–0	0.81 (0.51–1.31)	1.26 (0.89–1.79)	0.24 (0.06–0.90) *
	1–1 vs. 0–0	0.81 (0.50–1.31)	1.54 (1.07–2.22) *	0.81 (0.27–2.40)
Participates in Varsity Sports at School	0–1 vs. 0–0	1.38 (0.77–2.48)	1.06 (0.68–1.64)	0.51 (0.12–2.19)
	1–0 vs. 0–0	0.97 (0.58–1.62)	0.91 (0.62–1.34)	1.44 (0.39–5.25)
	1–1 vs. 0–0	1.46 (0.89–2.38)	0.79 (0.55–1.14)	1.14 (0.36–3.55)
Participates in Community Sports	0–1 vs. 0–0	2.31 (0.95–5.59)	1.27 (0.68–2.36)	1.08 (0.24–4.84)
	1–0 vs. 0–0	1.17 (0.76–1.82)	1.00 (0.73–1.36)	0.37 (0.13–1.07)
	1–1 vs. 0–0	2.08 (1.35–3.20) *	1.19 (0.86–1.63)	0.90 (0.35–2.36)
Parents Support Physical Activity	0–1 vs. 0–0	2.48 (1.07–5.75) *	0.99 (0.58–1.68)	0.79 (0.15–4.05)
	1–0 vs. 0–0	1.06 (0.53–2.14)	0.85 (0.54–1.34)	1.01 (0.26–3.85)
	1–1 vs. 0–0	2.13 (1.17–3.89) *	1.19 (0.81–1.76)	0.73 (0.24–2.28)
*p*-value of Hosmer–Lemeshow test		0.6035	0.4408	0.0788

Notes: models are adjusted for age and clustering by school. 0 = no, 1 = yes; 0–0 = no in grade 9, no in grade 12; 0–1 = no in grade 9, yes in grade 12; 1–0 = yes in grade 9, no in grade 12; 1–1 = yes in grade 9, yes in grade 12. ^†^ For current smoker, there were no respondents with the 1–0 response option for sweat, sleep, or sit, and no respondents with the 1–1 response option for sit. ^‡^ For current binge drinker, there were no respondents with the 1–0 or 1–1 response options for sit. * *p* < 0.05; CI, confidence interval.

**Table 6 ijerph-17-05326-t006:** Factors associated with meeting the 24-Hour Movement Guideline components in grade 12 among students who were not meeting the guideline in grade 9.

Parameter		Model 7: SWEAT *n*= 1601	Model 8: SLEEP *n* = 1112	Model 9: SIT *n* = 2252
		OR	OR	OR
Intercept		0.35 (0.00–35.29)	0.04 (0.00–3.00)	0.01 (0.00–21.55)
Sex	**Male**	2.05 (1.54–2.73) **	1.35 (1.00–1.82)	1.13 (0.72–1.78)
Current Smoker ^†^	0–1 vs. 0–0	1.31 (0.86–1.99)	1.14 (0.70–1.84)	0.54 (0.22–1.32)
	1–0 vs. 0–0	-	4.44 (0.19–101.60)	-
	1–1 vs. 0–0	1.29 (0.45–3.67)	1.48 (0.59–3.72)	-
Current Binge Drinker ^‡^	0–1 vs. 0–0	2.05 (1.43–2.92) **	0.84 (0.56-1.27)	0.66 (0.34–1.28)
	1–0 vs. 0–0	0.95 (0.2–4.62)	1.63 (0.58–4.57)	-
	1–1 vs. 0–0	0.75 (0.28–1.98)	1.33 (0.55–3.22)	1.64 (0.45–5.98)
Participates in Intramurals at School	0–1 vs. 0–0	0.99 (0.61–1.63)	0.80 (0.46–1.38)	0.57 (0.25–1.28)
	1–0 vs. 0–0	0.87 (0.56–1.36)	0.91 (0.58–1.43)	1.59 (0.86–2.94)
	1–1 vs. 0–0	1.20 (0.75–1.93)	0.99 (0.60–1.65)	0.57 (0.27–1.20)
Participates in Varsity Sports at School	0–1 vs. 0–0	1.05 (0.61–1.80)	1.20 (0.66–2.20)	1.80 (0.80–4.02)
	1–0 vs. 0–0	0.99 (0.60–1.64)	1.45 (0.89–2.38)	0.76 (0.33–1.72)
	1–1 vs. 0–0	1.19 (0.76–1.88)	1.34 (0.81–2.23)	1.55 (0.77–3.13)
Participates in Community Sports	0–1 vs. 0–0	2.78 (1.45–5.35) *	1.49 (0.74–2.99)	2.05 (0.73–5.72)
	1–0 vs. 0–0	1.65 (1.10–2.49) *	0.82 (0.53–1.26)	1.33 (0.69–2.56)
	1–1 vs. 0–0	2.84 (1.90–4.24) **	1.18 (0.78–1.78)	1.54 (0.81–2.93)
Parents Support Physical Activity	0–1 vs. 0–0	1.39 (0.75–2.59)	1.67 (0.90–3.10)	4.18 (1.31–13.32) *
	1–0 vs. 0–0	1.33 (0.77–2.30)	1.24 (0.71–2.17)	1.34 (0.39–4.55)
	1–1 vs. 0–0	1.08 (0.66–1.76)	1.25 (0.77–2.04)	2.49 (0.87–7.12)
*p*–value of Hosmer–Lemeshow test		0.9858	0.0232	0.9088

Notes: models are adjusted for age and clustering by school. 0 = no, 1 = yes; 0–0 = no in grade 9, no in grade 12; 0–1 = no in grade 9, yes in grade 12; 1–0 = yes in grade 9, no in grade 12; 1–1 = yes in grade 9, yes in grade 12. ^†^ For current smoker, there were no respondents with the 1–0 response option for sweat or sit, and no respondents with the 1–1 response option for sit. ^‡^ For current binge drinker, there were no respondents with the 1–0 response options for sit. * *p* < 0.05; ** *p* < 0.0001.

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
