# Peer review of "Factors Associated with Students Meeting Components of Canada’s New 24-Hour Movement Guidelines over Time in the COMPASS Study"

_ijerph, 2020, doi:10.3390/ijerph17155326_

Round 1

Reviewer 1 Report

The manuscript is well written and is focused on an important problem. There are minor suggestions to improve the readability of the paper:

line 34: replace "suspected" with "expected"

line 72-77: it seems that male sex is associated both with physical activity (line 72) and sedentary behavior (line 77). Please clarify.

line 104: "Ethics approval was received..." Is it possible that you meant "Ethics committee approval"? The sentence seems a bit awkward, as ethics is not likely to be approved or disapproved.

line 109: Please check. The sentence ends with "the".

For regression models, please report some measure of model fit, such as Nagelkerke pseudo r square. If the models explain very low proportion of variance, that could lead to another limitation of the study, as some important variables may be omitted. A short statement reporting on meeting the logistic regression assumptions may eliminate reasonable question of potential collinearity between predictors, such as organized sports and binge drinking.    

Author Response

The manuscript is well written and is focused on an important problem. There are minor suggestions to improve the readability of the paper:

RESPONSE: Thank you so much for your review of our manuscript. We really appreciate all your helpful feedback. We have addressed your comments/concerns throughout the manuscript and below you will find specific responses to each of your comments.

line 34: replace "suspected" with "expected"

RESPONSE: Thank you for catching this. The sentence was adjusted accordingly (line 34).

line 72-77: it seems that male sex is associated both with physical activity (line 72) and sedentary behavior (line 77). Please clarify.

RESPONSE: You are correct that these statements together are unclear. We have adjusted the sedentary behaviour predictor sentence (lines 81-82) and have added additional clarification (lines 82-84).  

line 104: "Ethics approval was received..." Is it possible that you meant "Ethics committee approval"? The sentence seems a bit awkward, as ethics is not likely to be approved or disapproved.

RESPONSE: This sentence was adjusted accordingly (line 114).

line 109: Please check. The sentence ends with "the".

RESPONSE: Thank you for catching this. The sentence was adjusted accordingly (line 119).

For regression models, please report some measure of model fit, such as Nagelkerke pseudo r square. If the models explain very low proportion of variance, that could lead to another limitation of the study, as some important variables may be omitted. A short statement reporting on meeting the logistic regression assumptions may eliminate reasonable question of potential collinearity between predictors, such as organized sports and binge drinking.

RESPONSE: We have included a statement in the methods to explain our testing for model fit (Hosmer-Lemeshow test) and the assumptions(line 212-215).

Please see attachment for revisions in the manuscript. 

Reviewer 2 Report

Thank you for the invite to review this manuscript. The study provides important insights into adolescents meeting the Canadian 24-MG and importantly, how this varies longitudinally and the predictors of these changes in behaviour. Some edits are required to improve the manuscript further, as detailed in the comments below:

Introduction

Line 34: For clarity, it would be beneficial to provide definitions of physical activity and sedentary behaviour so it is clear what distinct behaviours these represent

Line 36: Sleep should be introduced when discussing the ‘youth health behaviours’ as it seems out of place suddenly being included here without any introduction.

Lines 43-44: This sentence seems a little out of place, given the focus of the manuscript on adolescents

Lines 50-57: To clearly show how the SWEAT, STEP, SLEEP and SIT components link to the recommendations, please consider numbering these components to match with the subsequent numbered definitions.

Lines 63-67: Greater clarity is needed here with the use of the term ‘youth’. It is stated that ‘youth are not meeting the guidelines but it is not known how this evolve over secondary school’. But at what age does youth end? Does youth and secondary school overlap?

Line 68: As previously stated, a clear age range for the use of the term ‘youth’ is needed.

Line 88: For clarity, outlining the age of students at grade 9 should be included here

Materials and Methods

Line 109: This sentence is incomplete, please review.

Line 110: Please change ‘gender’ to ‘sex’ as this is the correct biological measurement term

Line 116: Please change ‘gender’ to ‘sex’ as this is the correct biological measurement term

Lines 123-124: If the Cq cannot measure light-PA, therefore the  STEP component of the 24-MG, how can the authors state ‘the Cq contains validated measures to determine if  participating students are meeting the 24-MG’ where they are not assessing all the components of these guidelines?

Line 124: It should be made clear that the inability to assess light-PA links to the STEP component of the 24-MG. Also, this is the first time the abbreviation of ‘light-PA’ has been used, so should be defined, or written in full

General: Where the measures of PA and SB and sleep collected at the same time of year at baseline and at follow-up? Seasonality may influence behaviours, so it should be stated if measurements were taken in the months. If not, this should be acknowledged as a limitation of the study

Lines 137-139: What was the rationale for picking these five different SBs? Since they are students, it seems strange to not have included time spent studying (i.e. when at school or homework) in this category as this is likely to be a highly sedentary activity.

Line 144: Please add a space between ‘2’ and ‘hours’

Line 150: Did smoking just consider smoking tobacco or other substances?

Line 165: Please change ‘physically activity’ to ‘physical activity’

Line 168: Please change ‘gender’ to ‘sex’ as this is the correct biological measurement term

Line 172: Please change ‘gender’ to ‘sex’ as this is the correct biological measurement term

Line 173: Please change ‘24MG’ to ’24-MG’ for consistency

General: In the analyses section, sometimes the models are referred to as MVPA, sleep etc. and sometimes they are referred to as the component of the 24-MG i.e. SWEAT, SLEEP, SIT. Please adopt the same approach throughout for consistency and clarity.

General: In this section please detail how data will be reported in the results section and what p-value was used to determine statistical significance

Results

Table 2: Please define as a footnote ‘OR’

Table 4: It is stated that McNemar’s test is used, however there is not mention of this is the analyses section of the methods; please add.

Discussion

Line 288: It should be made clear that this percentage refers to the Grade 9 sample statistic

Lines 296-298: Can the authors provide any references to support these presumptions?

Lines 352-355: Please provide a reference to support this statement. One may have been given as a superscript ‘4’, please reformat if this is to be used.

General: A large proportion of the discussion is focussed on the MVPA findings from this project. This section would be strengthened by the inclusion of some more discussion on the findings relating to SLEEP and SIT. For example, it was found community sport participation meant individuals were likely to sleep more. Why might this be the case? More time to recover from the physical activities, or more awareness of the need to recover after sport?

Thank you for the invite to review this manuscript. The study provides important insights into adolescents meeting the 24-MG and importantly, how this varies longitudinally and the predictors of this change in behaviour. Some edits are required to improve the manuscript further, as detailed in the comment below:

Introduction

Line 34: For clarity, it would be beneficial to provide definitions of physical activity and sedentary behaviour so it is clear what distinct behaviours these represent

Line 36: Sleep should be introduced when discussing the ‘youth health behaviours’ as it seems out of place suddenly being included here without any introduction.

Lines 43-44: This sentence seems a little out of place, given the focus of the manuscript on adolescents

Lines 50-57: To clearly show how the SWEAT, STEP, SLEEP and SIT components link to the recommendations, please consider numbering these components to match with the subsequent numbered definitions.

Lines 63-67: Greater clarity is needed here with the use of the term ‘youth’. It is stated that ‘youth are not meeting the guidelines but it is not known how this evolve over secondary school’. But at what age does youth end? Does youth and secondary school overlap?

Line 68: As previously stated, a clear age range for the use of the term ‘youth’ is needed.

Line 88: For clarity, outlining the age of students at grade 9 should be included here

Materials and Methods

Line 109: This sentence is incomplete, please review.

Line 110: Please change ‘gender’ to ‘sex’ as this is the correct biological measurement term

Line 116: Please change ‘gender’ to ‘sex’ as this is the correct biological measurement term

Lines 123-124: If the Cq cannot measure light-PA, therefore the  STEP component of the 24-MG, how can the authors state ‘the Cq contains validated measures to determine if  participating students are meeting the 24-MG’ where they are not assessing all the components of these guidelines?

Line 124: It should be made clear that the inability to assess light-PA links to the STEP component of the 24-MG. Also, this is the first time the abbreviation of ‘light-PA’ has been used, so should be defined, or written in full

General: Where the measures of PA and SB and sleep collected at the same time of year at baseline and at follow-up? Seasonality may influence behaviours, so it should be stated if measurements were taken in the months. If not, this should be acknowledged as a limitation of the study

Lines 137-139: What was the rationale for picking these five different SBs? Since they are students, it seems strange to not have included time spent studying (i.e. when at school or homework) in this category as this is likely to be a highly sedentary activity.

Line 144: Please add a space between ‘2’ and ‘hours’

Line 150: Did smoking just consider smoking tobacco or other substances?

Line 165: Please change ‘physically activity’ to ‘physical activity’

Line 168: Please change ‘gender’ to ‘sex’ as this is the correct biological measurement term

Line 172: Please change ‘gender’ to ‘sex’ as this is the correct biological measurement term

Line 173: Please change ‘24MG’ to ’24-MG’ for consistency

General: In the analyses section, sometimes the models are referred to as MVPA, sleep etc. and sometimes they are referred to as the component of the 24-MG i.e. SWEAT, SLEEP, SIT. Please adopt the same approach throughout for consistency and clarity.

General: In this section please detail how data will be reported in the results section and what p-value was used to determine statistical significance

Results

Table 2: Please define as a footnote ‘OR’

Table 4: It is stated that McNemar’s test is used, however there is not mention of this is the analyses section of the methods; please add.

Discussion

Line 288: It should be made clear that this percentage refers to the Grade 9 sample statistic

Lines 296-298: Can the authors provide any references to support these presumptions?

Lines 352-355: Please provide a reference to support this statement. One may have been given as a superscript ‘4’, please reformat if this is to be used.

General: A large proportion of the discussion is focussed on the MVPA findings from this project. This section would be strengthened by the inclusion of some more discussion on the findings relating to SLEEP and SIT. For example, it was found community sport participation meant individuals were likely to sleep more. Why might this be the case? More time to recover from the physical activities, or more awareness of the need to recover after sport?

Author Response

Response to Reviewer 2

Thank you for the invite to review this manuscript. The study provides important insights into adolescents meeting the Canadian 24-MG and importantly, how this varies longitudinally and the predictors of these changes in behaviour. Some edits are required to improve the manuscript further, as detailed in the comments below:

RESPONSE: Thank you so much for your thorough review of our manuscript. We really appreciate all your helpful feedback. We have addressed your comments/concerns throughout the manuscript and below you will find specific responses to each of your comments. Please see the attachment for revisions made in the manuscript. 

Introduction

Line 34: For clarity, it would be beneficial to provide definitions of physical activity and sedentary behaviour so it is clear what distinct behaviours these represent

RESPONSE: We have added a few examples of each behaviour in the introductory sentence (lines 33-34) and full definitions are provided in the following paragraph where the guidelines are detailed.

Line 36: Sleep should be introduced when discussing the ‘youth health behaviours’ as it seems out of place suddenly being included here without any introduction.

RESPONSE: Great point, we have adjusted to include sleep in the introductory statement (line 33).

Lines 43-44: This sentence seems a little out of place, given the focus of the manuscript on adolescents

RESPONSE:We have included this statement to demonstrate that 24-hour movement guidelines are becoming increasingly popular internationally, and are being expanded to cover additional age groups. We feel it provides important international context.

Lines 50-57: To clearly show how the SWEAT, STEP, SLEEP and SIT components link to the recommendations, please consider numbering these components to match with the subsequent numbered definitions.

RESPONSE: We have adjusted this accordingly (lines 53-54).

Lines 63-67: Greater clarity is needed here with the use of the term ‘youth’. It is stated that ‘youth are not meeting the guidelines but it is not known how this evolve over secondary school’. But at what age does youth end? Does youth and secondary school overlap?

RESPONSE: This is a great point. Youth does vary in definition depending on the study. We have adjusted the terminology throughout the introduction to be consistent with ages presented in each of the previous studies (lines 63;66;71;75;88).

Line 68: As previously stated, a clear age range for the use of the term ‘youth’ is needed.

RESPONSE: We have adjusted this accordingly in the introduction and have specified in the methods the age range associated with the term in this study (line 132).

Line 88: For clarity, outlining the age of students at grade 9 should be included here

RESPONSE: We have adjusted this accordingly (line 93).

Materials and Methods

Line 109: This sentence is incomplete, please review.

RESPONSE: Thank you for catching this. The sentence was adjusted accordingly (line 119).

Line 110: Please change ‘gender’ to ‘sex’ as this is the correct biological measurement term

RESPONSE: We have updated this throughout the manuscript.

Line 116: Please change ‘gender’ to ‘sex’ as this is the correct biological measurement term

RESPONSE: We have updated this throughout the manuscript.

Lines 123-124: If the Cq cannot measure light-PA, therefore the  STEP component of the 24-MG, how can the authors state ‘the Cq contains validated measures to determine if  participating students are meeting the 24-MG’ where they are not assessing all the components of these guidelines?

RESPONSE: You are correct, the Cq only has the capacity to measure the SWEAT, SLEEP and SIT components of the guidelines. The sentence was adjusted to accurately depict this (line 133-134).

Line 124: It should be made clear that the inability to assess light-PA links to the STEP component of the 24-MG. Also, this is the first time the abbreviation of ‘light-PA’ has been used, so should be defined, or written in full

RESPONSE: The sentence was adjusted accordingly (line 135).

General: Where the measures of PA and SB and sleep collected at the same time of year at baseline and at follow-up? Seasonality may influence behaviours, so it should be stated if measurements were taken in the months. If not, this should be acknowledged as a limitation of the study

RESPONSE:Yes, schools generally participate at the same time each year to minimize confounding from seasonal effects.

Lines 137-139: What was the rationale for picking these five different SBs? Since they are students, it seems strange to not have included time spent studying (i.e. when at school or homework) in this category as this is likely to be a highly sedentary activity.

RESPONSE:Consistent with the recommendation provided in the 24-hour guidelines, sedentary behaviour was defined as recreational screen time. We measured this through the 5 activities mentioned (watching/streaming TV, playing video games, talking on the phone, surfing the internet, and text messaging/emailing). We have clarified this in the methods (line 149). Although studying is technically a sedentary behaviour, it was left out of this analysis as we wanted to specifically examine recreational sedentary behaviours and do not want to discourage studying by suggesting that it is a harmful behaviour.

Line 144: Please add a space between ‘2’ and ‘hours’

RESPONSE: We have updated this throughout the manuscript (line 160).

Line 150: Did smoking just consider smoking tobacco or other substances?

RESPONSE: For the purpose of this manuscript, smoking was specific to smoking tobacco. We have clarified this in the manuscript (line 166).

Line 165: Please change ‘physically activity’ to ‘physical activity’

RESPONSE: Thank you for catching this, adjusted accordingly (line 182).

Line 168: Please change ‘gender’ to ‘sex’ as this is the correct biological measurement term

RESPONSE: We have updated this throughout the manuscript.

Line 172: Please change ‘gender’ to ‘sex’ as this is the correct biological measurement term

RESPONSE: We have updated this throughout the manuscript.

Line 173: Please change ‘24MG’ to ’24-MG’ for consistency

RESPONSE: We have updated this throughout the manuscript (line 190).

General: In the analyses section, sometimes the models are referred to as MVPA, sleep etc. and sometimes they are referred to as the component of the 24-MG i.e. SWEAT, SLEEP, SIT. Please adopt the same approach throughout for consistency and clarity.

RESPONSE: Have adjusted to be consistent throughout the manuscript (line 191).

General: In this section please detail how data will be reported in the results section and what p-value was used to determine statistical significance

RESPONSE: We have included a statement in the methods to detail the reporting and significance of results (line 215-217).

Results

Table 2: Please define as a footnote ‘OR’

RESPONSE: Have updated the table to improve clarity and included a footnote explaining OR and the 95% confidence interval (line 253).

Table 4: It is stated that McNemar’s test is used, however there is not mention of this is the analyses section of the methods; please add.

RESPONSE: Added the purpose of the test as well as the SAS procedure to the methods (line 196-198).

Discussion

Line 288: It should be made clear that this percentage refers to the Grade 9 sample statistic

RESPONSE:We have revised this sentence to include the reviewer’s comment (line 326).

Lines 296-298: Can the authors provide any references to support these presumptions?

RESPONSE: We have cited “Nusser, S.M.; Beyler, N.K.; Welk, G.J.; Carriquiry, A.L.; Fuller, W.A.; King, B.M.N. Modeling errors in physical activity recall data. J Phys Act Health2012; 9(Suppl 1), S56-S67. doi: https://doi.org/10.1123/jpah.9.s1.s56” to support this statement. (line 329).

Lines 352-355: Please provide a reference to support this statement. One may have been given as a superscript ‘4’, please reformat if this is to be used.

RESPONSE: You are correct, we accidentally did not reformat this citation. Adjusted to match formatting requirements (line 411)

General: A large proportion of the discussion is focussed on the MVPA findings from this project. This section would be strengthened by the inclusion of some more discussion on the findings relating to SLEEP and SIT. For example, it was found community sport participation meant individuals were likely to sleep more. Why might this be the case? More time to recover from the physical activities, or more awareness of the need to recover after sport?

RESPONSE: Thank you for raising this point, you are correct that we should have discussed this finding in more detail. We have added some discussion around this concept (lines 356-361). Interestingly the MVPA recommendation saw the greatest number of significant results and so it did guide the majority of the discussion. Although the heading was missing, we tried to provide a high level summary of the results across all recommendations in the conclusion.
